# Comparison of Biochemical Composition and Non-Volatile Taste Active Compounds of Back and Abdominal Muscles in Three Marine Perciform Fishes, *Chromileptes altivelis*, *Epinephelus akaara* and *Acanthopagrus schlegelii*

**DOI:** 10.3390/molecules27144480

**Published:** 2022-07-13

**Authors:** Yun Sun, Guisen Chen, Zhenjie Cao, Chunsheng Liu

**Affiliations:** 1State Key Laboratory of Marine Resource Utilization in South China Sea, Hainan University, Haikou 570228, China; syshui207@126.com; 2Ocean College, Hainan University, Haikou 570228, China; thecgs001@foxmail.com (G.C.); 993674@hainanu.edu.cn (Z.C.)

**Keywords:** perciform, fatty acid, non-volatile taste active compounds, equivalent umami concentration (EUC)

## Abstract

Humpback grouper *Chromileptes altivelis* (HG), red-spotted grouper *Epinephelus akaara* (RG) and black seabream *Acanthopagrus schlegelii* (BS) are three popular perciform fishes with an increasingly important farming industry. The prices of BS are much lower than other grouper species; however, the differences in the nutritive values of these three perciform fishes with commercial specifications have not been reported. In this study, the biochemical composition and non-volatile taste active compounds of adult HG, RG and BS were investigated. Moisture contents in BS were significantly higher than in HG and RG (*p* < 0.05), and relatively lower crude protein contents in BS were observed. Lipid contents of back muscle were lower than that of abdomen muscle in the three fish species. C22:6n-3 (DHA) was the major poly-unsaturated fatty acid (PUFA) in HG and BS, while the main PUFA in RG was C18:2n-6. The total healthy omega-3 fatty acid (Σn-3) profiles in HG were the highest (24.08–24.59%), followed by RG (18.24–19.06%) and BS (13.63–15.91%) (*p* < 0.05). Glycine was the most abundant free amino acid (FAA) in HG and RG, while lysine was the major FAA in BS. Equivalent umami concentration (EUC) values in BS were the highest, followed by HG and RG (*p* < 0.05). Lactic acid and PO_4_^3−^ were the major organic acids and inorganic ions, respectively. In conclusion, HG and RG provided more protein and healthy omega-3 fatty acids than BS, while BS had a stronger umami taste according to the EUC values.

## 1. Introduction

Seafood, including fish and shellfish, is considered as part of a healthy and balanced diet [1,2]. The total global aquaculture production of fish and shellfish reached more than 80 Mt in 2017, and the global per capita seafood supply was estimated at 10.5 kg [2]. Perciform (order Perciformes) is the largest group of fishes in the world, including more than 6000 species placed in about 150 families. Most perciform species are marine fishes, generally living along coastal areas of tropical and temperate regions, and the group plays an important role in world fisheries [3]. Most marine perciforms tend to be carnivorous such as groupers, snappers, pompano and tuna, which are cultured and supply high quality meat for people around the world [1,3,4].

Humpback grouper (*Chromileptes altivelis*, abbreviated as HG), red-spotted grouper (*Epinephelus akaara*, abbreviated as RG) and black seabream (*Acanthopagrus schlegelii*, abbreviated as BS) are three popular perciform fishes with an increasingly important farming industry [5,6,7]. In Southeast Asia and China, HG and RG are considered as premium seafood sold at high prices in live seafood markets, at about 700 and 500 renminbi (RMB)/kg, respectively, while black seabream was about 50 RMB/kg, which is much lower than other grouper species [6]. Commercially, the resource output, breeding cost, quality in nutrition and taste are the key factors accounted for in the fish price. For example, HG and RG both live in coral reefs, and the resources of these two grouper species are becoming rare, and they are even listed as endangered species in the International Union for Conservation of Nature (IUCN) Red List [8]. Furthermore, the high cost of these two fishes is also attributed to their lower growth rate and higher culture technical requirements. On the contrary, BS is a very popular aquaculture marine species in China, Korea, Japan and many Southeast Asian countries [5,9].

Until now, many studies have focused on the effects of different dietary sources on the growth performance and biochemical composition of juvenile HG, RG and BS [5,10,11,12]. For example, Shapawi et al. [13] analyzed the fatty acid composition of juvenile HG (∼10.6 g) after feeding with diets containing different vegetable oils; Wang et al. [14] determined the effects of dietary protein and lipid levels on the growth performance and body composition of juvenile RG (∼7.85 g); and Jin et al. [12] reported the effects of different dietary lipid sources on the physiological characteristics and lipid metabolism in juvenile BS (∼5.4 g). As for the quality differences in nutrition and taste, only a few studies exist on the nutritional composition of these three perciform fishes with commercial specification at 0.6–0.8 kg [15], while no reports have been reported on the flavor compounds of these three fish species.

Fish meat is considered as a low-fat and protein-rich source, with other nutritional components that are positively attributed to health. Its nutritional quality is measured by the composition of amino acids, healthy omega-3 long chain polyunsaturated fatty acids, such as eicosapentaenoic acid (EPA) and docosahexaenoic acid (DHA), and other micronutrients, e.g., iodine, zinc, magnesium and calcium [1]. In addition, the volatile and non-volatile profiles are also important characteristics that influence the flavor of fish meat and consumers’ preference and acceptance [16,17]. For example, umami is a specific taste of seafood, mainly attributed to umami amino acids and 5′-nucleotides; organic acids provide the acidic taste and other flavors, while inorganic ions are essential auxiliary flavor components in seafood [18,19]. In this study, to provide and compare nutritional and flavor information of HG, RG and BS with commercial specification, the biochemical composition and non-volatile compounds of both back and abdominal muscles in these three perciform fishes were investigated.

## 2. Results and Discussion

### 2.1. Proximate Compositions

Table 1 represents the proximate composition (% of wet weight) of three perciform fishes in this study. The moisture values of BS in back and abdomen muscles were 76.13% and 76.40%, respectively, which were significantly higher than that in HG and RG (*p* < 0.05). Crude protein contents in HG and RG were slightly higher than that in BS, and significant differences were observed in HG and RG back muscles vs. BS abdomen muscle (*p* < 0.05). The crude lipid contents of back muscles were lower than that of the abdomen in each fish species, and significant differences were observed between back and abdomen muscles of RG and BS (*p* < 0.05). Ash contents in the three fish species ranged from 1.58 to 1.92% without any significant differences.

The moisture content of fish is influenced by many factors, such as species, diet, age, and rearing regions [5,20]. In a previous study, the moisture contents of juvenile HG, RG and BS were 68.7–71.5%, 75.38% and 75.60–76.42%, respectively, which are similar to the results in our experiment [5,9,10]. In this study, RG and BS were cultured in the same province—Fujian province—while HG was reared in Hainan province. Therefore, species might be the main factor behind moisture differences among species. As for the significant differences of the level of crude lipid in back and abdominal muscles of the three perciform fishes, they might be ascribed to the functional differences in back and abdominal muscles. A higher lipid content in abdominal muscles has also been reported in freshwater fishes [21].

### 2.2. Fatty Acid Composition

The fatty acid profiles of the three perciform fishes are shown in Table 2. In this study, C16:0 (18.94–24.90%), C18:1n-7 (15.56–29.76%) and C22:6n-3 (docosahexaenoic acid, DHA, 12.60–19.73% for HG and RG)/C18:2n-6 (20.37–21.23% for BS) were the major saturated fatty acids (SFA), mono-unsaturated fatty acids (MUFA) and poly-unsaturated fatty acids (PUFA), respectively. These results were in accordance with other studies of these three fish species. For example, the major SFA, MUFA and PUFA for juvenile HG were C16:0, C18:1n-9 and C22:6n-3 [10]; for young RG they were C16:0, C18:1 and C22:6n-3 [11]; and for juvenile BS they were C16:0, C18:1n-9 and C18:2n-6, respectively [12]. Besides species, the different dietary lipid sources were also a key factor influencing the fatty acid profiles, especially the PUFA, of fish [22]. Jin et al. [12] reported that the C18:2n-6 and C22:6n-3 compositions in the perilla-oil diet group of juvenile BS were 14.3% and 3.12%, respectively, while the profiles of these two PUFAs changed to 7.33% and 19.45%, respectively, in the EPA + DHA diet group. In HG muscle, the profile of C22:6n-3 also varied from 10.2% in the cod liver oil-feeding group to 4.0% in the crude palm oil-feeding group [10]. In other perciform fishes, such as silver pomfret (*Pampus argenteus*), sardine (*Sardinops melanostictus*) and large yellow croaker (*Larimichthys crocea*), the fatty acid compositions also showed a clear relationship with daily diet [23,24,25].

Though there were almost no significant differences in fatty acid compositions of back and abdominal muscles in the same fish species, significant differences in fatty acid profiles were observed among different fish species (*p* < 0.05). For example, the profiles of C16:0, C18:0 and C22:6n-3 were highest in HG compared with that in RG and BS (*p* < 0.05); the profiles of C18:1n-7 and C16:1n-7 in BS were 26.30–29.76% and 6.53–7.28%, respectively, which were significantly higher than that in HG and RG (*p* < 0.05); the profiles of C18:2n-6 in RG were 20.37–21.23%, which were significantly higher than that in HG and BS (*p* < 0.05). Furthermore, the levels of PUFA, Σn-6 and Σn-3 varied among the species. The Σn-6 contents in RG was the highest (21.99–22.87%), followed by BS (8.98–10.50%) and HG (6.55–6.78%), while the profiles of Σn-3 were HG (24.08–24.59%) > RG (18.24–9.06%) > BS (13.63–15.91%). Accordingly, the ratio of Σn-3/Σn-6 in HG, BS and RG showed an obvious decline.

It is notable that fish are the main source of omega-3 long-chain PUFA (>C18), such as DHA and eicosapentaenoic acid (EPA, C20:5n-3) for human consumption, which may reduce the risk of heart disease and promote brain and eye health [4]. In this study, the DHA was the major PUFA in HG and BS, which showed the same result as other sea fish, such as yellow drum (*Spotted maigre*) and small yellow croaker (*L**arimichthys*
*polyactis*) [25,26], while in some other fish, such as large yellow croaker and hybrid grouper (*Epinephelus lanceolatus* ♂ × *Epinephelus fuscoguttatus* ♀), the major PUFA is C18:2n-6 [27,28]. In studies by Twining et al. [29] and Golden et al. [4] of domesticated terrestrial animals, the high contents of C18:2n-6 was ascribed to the high content of short chain ω-3 fatty acids in their terrestrial plants diet.

Furthermore, the Σn-3 profile in HG was the highest, followed by RG and BS, which meant HG had a more excellent fatty acid compositions compared to the other two fish species. The Σn-3/Σn-6 ratio recommended by the World Health Organization is 1:1 or above [30]. In this study, the contents of Σn-3/Σn-6 in HG and BS were more than 1, while the value of RG was 0.80–0.87, which was attributed to the high composition of C18:2n-6 (20.37–21.23%). Toledo et al. [31] also detected a higher 18:2n-6 content (4.89–20.05%) in cultured RG muscle, compared to the relatively lower 18:2n-6 content (0.36–1.78%) in wild samples. C18:2n-6 was the main PUFA in plants; therefore, the higher contents of plant-based food in RG’s daily diet might be one of the reasons behind the high C18:2n-6 content [30].

### 2.3. Comparison of Free Amino Acids (FAAs) and 5′-Nucleotides

In this study, seventeen FAAs were identified in three perciform fishes (Table 3). Of these, glycine was the most abundant in HG and RG (53.26–86.30 mg/g of wet weight), followed by alanine and lysine, while lysine was the major FAA in BS (46.82–55.37 mg/g of wet weight), followed by glycine and alanine. The differences in the FAA contents could be related to different actors, such as species, type of diet or rearing condition [18]. The contents of glycine in HG and RG were almost the same as that in large yellow croaker and hybrid grouper [27,28], while the high lysine contents in FAA were also observed in gilthead sea bream (*Sparus aurata*), which was the same as that in BS [32]. In other fish species, the major FAAs varied. For example, the major FAA in fingerling rainbow trout (*Oncorhynchus mykiss*), sardine and Atlantic salmon (*Salmo salar*) were glycine, histidine and arginine, respectively, and protein levels and season changes significantly affected the composition of FAA [23,33,34].

FAAs are related to the characteristic flavor of fish, such as umami, sweetness and bitterness [27,34]. The total FAAs in back and abdomen muscles of BS were 301.59 and 271.03 mg/100 g of wet weight, which was significantly higher than that in the two other fish species (*p* < 0.05). Correspondingly, the contents of MSG-like, bitter and tasteless FAAs of BS were significantly higher compared to the two grouper species (*p* < 0.05). On the contrary, the sweet FAAs in RG and HG were significantly higher than that in BS (*p* < 0.05), except of abdomen muscle in HG. FAA in fish plays essential roles in metabolism, such as the adjustment of osmotic pressure and as an energy source [23]. BS belongs to eurythermal and euryhaline fish species, and therefore, a higher FAA contents in muscle could enhance the adaption of BS to a changing environment. The relatively higher sweet FAAs in RG and HG, which were mostly caused by glycine, might be ascribed to different environmental adaptabilities between groupers and BS.

As shown in Table 4, 5′-inosine monophosphate (IMP) was the main 5′-nucleotide, followed by 5′-adenosine monophosphate (AMP) and 5′-guanosine monophosphate (GMP) in the three fish species. There were almost no significant differences in IMP contents among different species and muscle parts (*p* < 0.05), except for back muscle in BS. The AMP content was the highest in HG, followed by BS and RG, and significant differences were observed among the three fish species (*p* < 0.05). In a previous study, IMP was found to be the major flavor-contributing 5′-nucleotide in many fish species, such as sea bream, rainbow trout and tilapia *(Oreochromis niloticus),* and other species, such as oyster *Crassostrea hongkongensis* [18,35,36]. IMP is responsible for taste complexity and sweetness, which could contribute to a very pleasant taste, at a low concentration, while AMP is responsible only for sweetness at concentrations from 50 to 100 mg/100 g [19]. In this study, the AMP contents of the three fish were in the range of 0.91–12.22 mg/100 g, and were much lower than the taste threshold. Therefore, IMP was the major 5′-nucleotide contributing to taste.

### 2.4. Comparison of Malic and Lactic Acid Contents

As shown in Table 5, two organic acids including malic and lactic acid were detected in muscles of the three fish species. The malic acid contents in HG, RG and BS were 0.45–0.59, 0.19–0.33 and 0.17–0.18 mg/g of wet weight, respectively, which showed a significantly decreased tendency (*p* < 0.05). Significant differences were observed in back and abdomen muscles of the same fish species in HG and RG (*p* < 0.05). Furthermore, the contents of lactic acid in BS were the highest (5.26–7.95 mg/g of wet weight), followed by RG (4.90–5.26 mg/g of wet weight) and HG (3.18–3.44 mg/g of wet weight), and significant differences were found among species (*p* < 0.05).

In seafood products, malic, lactic, citric and succinic acid are the main taste-activating organic acids [18]. These compounds were found to have significant differences in composition and concentrations according to species [18,19,25,37]. For example, lactic, tartaric, malic and succinic acids were detected in yellowfin tuna (*Thunnus albacores*), and lactic acid was the major organic acid (4.98 g/100 g) [37]. In pufferfish (*Takifugu obscurus* and *T. rubripes*) and rainbow trout, lactic was the major organic acid [36,37]. Similarly, lactic acid was found to be the main acid in squids [19]. In our study, lactic and malic acids were detected in three perciform fishes. However, the malic acids content in these three fish were lower than its taste activity values (TAVs) (50 mg/100 g), except of the abdomen muscle of HG. Lactic acid could provide a sour and umami taste, and its contents in these three fish were higher than its TAV (126 mg/100 g), which means lactic acid is a major acidic component responsible for the taste of these three fishes [18]. Organic acids are closely related to the synthesis and metabolism of aromatic compounds, amino acids and esters [17]. The differences in the lactic acid in three fish species may be caused by the differences in metabolism in different fish species.

### 2.5. Comparison of Inorganic Ions

Inorganic ions in different muscle portions of three fish species are shown in Table 5. Based on wet weight, PO_4_^3−^ (4.86–7.01 mg/g of wet weight) and K^+^ (3.93–4.79 mg/g of wet weight) were the most abundant inorganic ions in the three fish species, followed by Cl^−^ (0.53–1.34 mg/g of wet weight), Na^+^ (0.47–1.09 mg/g of wet weight), Ca^2+^ (0.37–0.54 mg/g of wet weight) and Mg^2+^ (0.30–0.34 mg/g of wet weight). In detail, the PO_4_^3−^ in BS was highest, followed by RG and HG, and its contents in back muscle of the same fish species were significantly higher than that in abdomen muscle (*p* < 0.05), except of BS. The Cl^−^ in back and abdomen muscles of BS were 0.78 and 1.34 mg/g of wet weight, respectively, which was significantly higher than that in HG and RG (*p* < 0.05), except for BS back muscle vs. RG abdomen muscle. Furthermore, the Na^+^ in BS was also higher than in the other two species, and significant differences were observed between BS back muscle and other groups (*p* < 0.05).

Inorganic ions are an essential auxiliary flavor component in seafood [18]. Na^+^ contributes a strong salty taste, while K^+^ provides a bitter and salty taste. In our study, K^+^ was the most abundant cation, which meant this mineral contributed to the meat flavor in these three fish species. Similar phenomena were also reported in muscles of other marine fish, such as large yellow croaker [20] and pufferfish [37]. Furthermore, PO_4_^3−^ could reduce bitterness and increase the intensities of umami tastes and sourness, and Cl^−^ was found to suppress the sour taste and enhance umami taste and sweetness [19]. In our experiment, the contents of PO_4_^3−^ were much higher than that of Cl^−^, which showed the same tendency with Yangtze (*Coilia ectenes*) and squid [19,38], while the opposite result was observed for steamed pufferfish [39]. The differences in the inorganic ions in seafood products were probably caused by species, reproductive status, age, sex, or assay methods (for example, mercury thiocyanate and molybdemum blue methods were used in Cl^−^ and PO_4_^3−^ were detection in some studies) [19,40].

### 2.6. Comparison of Equivalent Umami Concentration (EUC) Values

The EUC values of three perciform fishes were calculated according to the equation used to evaluate the results of sensory evaluation (Figure 1). The value was highest in BS back muscle, followed by its abdomen muscle, HG back and abdomen muscles, and RG back and abdomen muscles (*p* < 0.05). Though the EUC value has its limitations, it is considered as a key indicator in evaluating the umami taste of seafood, such as fish, squid and oyster [18,19,38]. In our study, glutamic acid and IMP were the main flavor compositions in the three perciform fishes. Besides the EUC value, umami was also related to other factors, such as inorganic components [19]. For example, Na^+^ can improve the overall umami taste; Cl^−^ and PO_4_^3^^−^ could increase the intensities of monosodium glutamate (MSG)-like tastes; and Ca^2+^ showed a negative correlation with EUC [18]. Therefore, the umami taste of seafood was made up of multiple flavours, and not only limited to umami FAA and 5′-nucleotide, but also some inorganic ions.

## 3. Materials and Methods

### 3.1. Collection and Preparation of Three Marine Perciform Fishes

Live HG *C. altivelis* (633.13 ± 48.37 kg of wet body weight), RG *E. akaara* (704.23 ± 27.17 kg of wet body weight) and BS *A. schlegelii* (718.48.33 ± 38.16 kg of wet body weight) were purchased from commercial fish farms in Danzhou, Chaozhou and Ningde, China, respectively. Nine fish of each species were sampled, and randomly divided into three groups. Fish were euthanized with tricaine methanesulfonate (Sigma-Aldrich, St. Louis, MO, USA), and then back and abdominal muscles of each fish were carefully collected. Each part within one group was mixed together and stored at −80 °C for subsequent chemical analysis.

### 3.2. Proximate Analysis

The proximate composition of muscles was analyzed according to the method described by Liu et al. [18]. The moisture was determined by oven drying the sample at 60 °C until a constant weight was achieved. For the determination of other proximate compositions, samples were subjected to freeze drying at −60 °C in a vacuum freeze dryer (SCIENTZ-10 N, Ningbo, China), and then grinded into powder. Crude protein content was determined with the Kjeldahl method using a fully automated Kjeldahl nitrogen/protein analyzer (FOSS-Soxtec 2050, Höganäs, Sweden) after acid digestion in a Tecator digestor 2020 [41]. Crude lipid was extracted with petroleum ether and crude ash was determined gravimetrically in a muffle furnace by incineration at 550 °C for 24 h. All samples were analyzed in triplicate.

### 3.3. Fatty Acid Analysis

The fatty acids were extracted and fatty acid methyl esters (FAMEs) were prepared according to previous studies [18,25]. In detail, total lipids were extracted from 2 g of homogenised fish meat with chloroform–methanol (2:1, *v/v*), then saponified, followed by esterification, and finally the extraction of FAMEs in hexane. The FAMEs were analyzed using a Gas Chromatography-Mass Spectrometer (GC-MS) (7894A-5975C; Agilent, Santa Clara, CA, USA). The GC and MS parameters were as follows: sample volume, 1 μL; split injection, 2:1; injector temperature, 260 °C; column, HP-5MS (30 m × 250 μm × 0.25 μm, Agilent); carrier gas, helium (99.999%); flow rate, 3.0 mL/min; transfer line temperature, 280 °C; ion source temperature, 230 °C; and ionisation energy, −70 eV. The oven temperature was programmed from 50 °C to 200 °C at 5 °C/min, then to 230 °C at 2 °C/min and held for 10 min. Mass spectra were scanned from 50 to 550 amu in the total ion chromatogram (TIC) mode to identify the various compounds. Fatty acids were identified by comparing the retention times of FAME with the Supelco^TM^ 37 component FAME mixture (Cat. No. 47885-U; Supelco, Yuanye Biotechnology Co. Ltd, Shanghai, China). Quantitative data were calculated using the peak area ratio (% total fatty acids). All measurements were performed in triplicate.

### 3.4. Free Amino Acid (FAA) Analysis

FAAs were performed as described previously [18]. In detail, a 2.5 g sample was homogenized in three volumes of 10% trichloroacetic acid (TCA), and centrifuged at 10,000× *g* for 15 min at 4 °C. Supernatants were separated into 25-μL aliquots. Samples were then analyzed for FAAs using a Waters 2996 high-performance liquid chromatograph (HPLC) (Waters Corporation, Milford, MA, USA) equipped with a Waters 2996 Photodiode Array Detector. Samples (5 μL) were separated using a Waters Pico-Tag-C18 column (3.9 mm × 150 mm). The mobile phase consisted of AccQ-Tag Elent A (*v/v* = 10:1) (A), 100% acetonitrile (B) and deionized water (C). Gradient elution was performed as follows: 100% A to 99% A and 1% B at 0.5 min, to 94% A and 6% B at 18 min, to 90% A and 10% B at 19 min, to 83% A and 17% B at 29.5 min, to 60% B and 40% C at 33 min, to 100% A at 36 min, and to 100% A at 45 min. The detection wavelength was 248 nm. All analyses were repeated in triplicate. The identity and quantity of each amino acid was assessed by comparing the retention times and peak areas of each amino acid standard (Sigma-Aldrich, St. Louis, MO, USA).

### 3.5. 5′-Nucleotide Assay

Nucleic acid-related compounds were analyzed according to the previously described method [18]. The sample (~5 g) was homogenized in 25 mL of 0.6 M perchloric acid (HClO_4_) for 15 min, and centrifuged at 10,000× *g* for 10 min at 4 °C. The supernatants were collected and neutralized with 1 M potassium hydroxide (KOH). Next, sample solutions were filtered through a 0.45-μm cellulose membrane before HPLC analysis. A 10-μL filtered sample was injected into the HPLC (Waters 2996 HPLC, Waters, Milford, MA, USA) equipped with an Intersil ODS-3 column (4.6 mm × 250 mm, 5 μm) (Shimadzu (China) Ltd., Shanghai, China). The HPLC conditions were as follows: the temperature was 30 °C; eluents A and B were methanol and 0.05% phosphoric acid, respectively; the flow rate was 1.0 mL/min; and the UV detection wavelength was 260 nm. All samples were assayed in triplicate. Each nucleotide was identified and quantified with standard curves (Sigma-Aldrich, St. Louis, MO, USA).

### 3.6. Equivalent Umami Concentration (EUC)

The equivalent umami concentration (EUC, g MSG per 100 g of material weight) was the concentration of MSG equivalent to the umami intensity as determined by the mixture of MSG-like amino acids (Asp and Glu) and 5′-nucleotides (IMP, GMP and AMP). The EUC is represented by the following equation:Y = Σaibi + 1218(Σaibi) (Σajbj)(1)
where Y equals g MSG per 100 g wet sample; a_i_ is the concentration (g/100 g) of each umami amino acid; b_i_ is the relative umami concentration (RUC) for each umami MSG-like amino acid (Glu, 1 and Asp, 0.077), a_j_ is 5′-nucleotide concentration (g/100 g), b_j_ is the RUC for each umami 5′-nucleotide to IMP (IMP, 1; GMP, 2.3 and AMP, 0.18) and 1218 is a synergistic constant [18].

### 3.7. Malic, Lactic, Citric and Succinic Acids Assay

The sample (~1 g) was homogenized in 5 mL of purified water for 5 min, and then centrifuged at 10,000× *g* for 20 min at 4 °C. The supernatants were filtered through a 0.45-μm cellulose membrane prior to HPLC analysis. The HPLC conditions were the same as those used for 5′-nucleotide detection, except the detector wavelength of 215 nm. All samples were detected in triplicate. Each organic acid, including malic, lactic, citric and succinic acid, was identified and quantified using known standard curves (Sinopharm Chemical Reagent Co. Ltd., Shanghai, China).

### 3.8. Inorganic Ions Assay

The concentrations of K, Na, Ca, Mg, Mn and Zn in muscles were measured using an inductively coupled plasma optical emission spectrometer (OPTIMA-7000DV, Perkin-Elmer, Waltham, MA, USA) according to the method described by Yue et al. [19]. Approximately 0.5 g of the dried sample was digested with a reagent mixture of 1 mL of H_2_O_2_ and 5 mL of HNO_3_, and was then submitted to a heating program in a closed microwave oven with the following steps: 1 min at 320 W, 2 min at 0 W, 5 min at 320 W, 5 min at 520 W and 5 min at 740 W. The digested solution was diluted with deionized water to 100 mL and analyzed using an inductively coupled plasma optical emission spectrometer.

The phosphate and chloride concentrations were determined using an 882 ion chromatograph system (Metrohm, Riverview, FL, USA) equipped with a high capacity anion exchange analytical column (Metrosep A Supp 5, 250 mm × 4.0 mm, Metrohm, Riverview, FL, USA). The sample (1.5 g) was ashed in a muffle furnace at 550 °C for 24 h. After dissolving in deionised water, it was transferred to a 100-mL volumetric flask, then diluted with deionized water to the desired volume and mixed. The electrical conductivity of the sample solution was adjusted to 350 µS/cm with ultrapure water. Then, the sample solution was filtered through a 0.22 µm Durapore membrane (Millipore, Billerica, MA, USA) and injected into the ion chromatography system. All experiments were performed at room temperature at a flow rate of 0.7 mL min^−1^. The isocratic elution was carried out using a mixture of 3.2 mM Na_2_CO_3_ and 1.0 mM NaHCO_3_. All samples were detected in triplicate.

### 3.9. Statistical Analysis

Statistical treatment of the data was performed using the Data Processing System (DPS) statistical software. Data of biochemical composition, volatile and non-volatile compounds of all groups were analyzed by conducting a two-way analysis of variance (ANOVA), and means were subsequently separated using Tukey′s test. Prior to ANOVA, homogeneity of variances was tested using Levene′s Test. The results are presented as means ± SD.

## 4. Conclusions

The biochemical composition and non-volatile taste active compounds of back and abdominal muscles in three marine perciform fishes were compared. The crude protein contents in HG and RG were significantly higher than in BS (*p* < 0.05). Though there were no significant differences in lipid contents, the profiles of healthy omega-3 fatty acids in HG were the highest, followed by RG and BS. According to the EUC values, BS back muscle showed the best flavor taste, followed by its abdomen muscle, HG back and abdomen muscles, and RG back and abdomen muscles (*p* < 0.05). These findings clearly showed the differences in nutritional composition and non-volatile compounds of three adult perciform fishes, in which HG could provide higher levels of healthy omega-3, and BS could provide a stronger umami flavor taste for humans. Furthermore, besides the above parameters, other quality traits, such as texture characteristics, surface color and volatile flavor are also important factors for consumers’ acceptability of fish meat that need to be studied further.

## Figures and Tables

**Figure 1 molecules-27-04480-f001:**
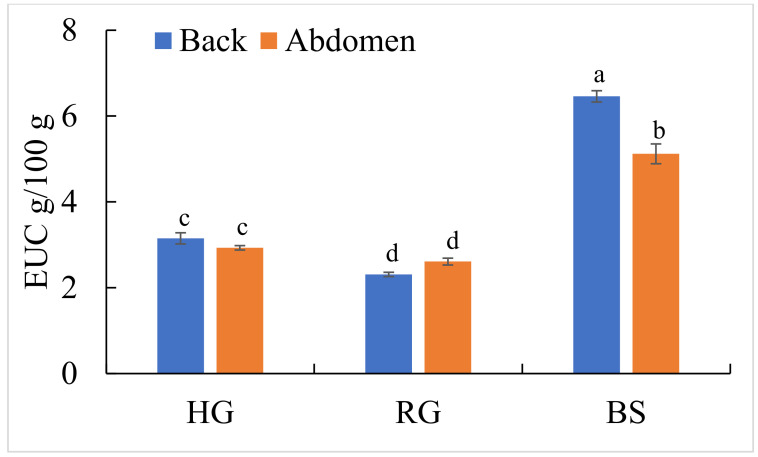
The equivalent umami concentration (EUC, g MSG/100 g of wet weight) of back and abdominal muscles in three perciform fishes. Different letters indicate significant difference (*p* < 0.05).

**Table 1 molecules-27-04480-t001:** Proximate composition of back and abdominal muscles in three perciform fishes.

Compositions (% of Wet Weight)	Humpback Grouper		Red-Spotted Grouper		Black Seabream	
	Back	Abdomen	Back	Abdomen	Back	Abdomen
Moisture	73.17 ± 1.13 b	72.81 ± 1.29 b	74.09 ± 0.88 b	73.36 ± 1.03 b	76.13 ± 0.85 a	76.40 ± 1.02 a
Crude protein	21.42 ± 0.26 a	21.31 ± 0.67 a	21.06 ± 0.75 a	20.15 ± 0.33 a	19.24 ± 0.75 ab	17.93 ± 0.80 b
Crude lipid	3.29 ± 0.22 ab	4.10 ± 0.24 a	2.82 ± 0.12 bc	3.98 ± 0.13 a	2.38 ± 0.31 c	3.35 ± 0.27 ab
Ash	1.92 ± 0.12	1.59 ± 0.21	1.67 ± 0.32	1.68 ± 0.16	1.72 ± 0.10	1.58 ± 0.17

Different letters within a row indicate significant difference (*p* < 0.05).

**Table 2 molecules-27-04480-t002:** Fatty acid profiles (%) of back and abdominal muscles in three perciforms fishes.

	Humpback Grouper		Red-Spotted Grouper		Black Seabream	
	Back	Abdomen	Back	Abdomen	Back	Abdomen
Saturated fatty acids (SFA)						
C12:0	0.06 ± 0.00	0.07 ± 0.00	0.05 ± 0.00	0.05 ± 0.00	0.07 ± 0.01	0.07 ± 0.02
C13:0	0.07 ± 0.01	0.06 ± 0.01	0.09 ± 0.01	0.05 ± 0.03	0.08 ± 0.02	0.05 ± 0.02
C14:0	3.61 ± 0.11 ab	4.02 ± 0.30 a	3.36 ± 0.31 bc	3.97 ± 0.17 a	2.60 ± 0.11 d	3.12 ± 0.09 c
C15:0	0.68 ± 0.04 a	0.76 ± 0.04 a	0.50 ± 0.00 b	0.53 ± 0.03 b	0.38 ± 0.05 c	0.40 ± 0.03 c
C16:0	24.90 ± 0.77 a	25.26 ± 0.45 a	20.80 ± 0.47 b	18.94 ± 0.58 c	22.41 ± 0.22 b	21.56 ± 0.85 b
C17:0	1.05 ± 0.06 a	1.11 ± 0.01 a	0.44 ± 0.02 b	0.41 ± 0.03 b	0.51 ± 0.03 b	0.52 ± 0.05 b
C18:0	10.90 ± 0.35 a	10.85 ± 0.25 a	5.67 ± 0.57 c	4.37 ± 0.14 d	7.35 ± 0.29 b	6.99 ± 0.77 b
C20:0	0.32 ± 0.02	0.36 ± 0.03	0.27 ± 0.04	0.24 ± 0.02	0.29 ± 0.04	0.34 ± 0.03
ΣSFA	41.58 ± 1.27 a	42.49 ± 1.07 a	31.18 ± 1.39 bc	28.56 ± 0.97 c	33.69 ± 0.76 b	33.05 ± 1.64 b
Mono-unsaturated fatty acids (MUFAs)						
C23:0	1.18 ± 0.10 a	1.19 ± 0.08 a	0.54 ± 0.01 c	0.39 ± 0.03 c	0.74 ± 0.18 b	0.72 ± 0.03 b
C14:1n-3	0.06 ± 0.01	0.05 ± 0.01	0.09 ± 0.03	0.05 ± 0.02	0.09 ± 0.01	0.06 ± 0.01
C14:1n-5	0.09 ± 0.01 b	0.08 ± 0.03 b	0.11 ± 0.01 b	0.08 ± 0.02 b	0.18 ± 0.01 a	0.17 ± 0.03 a
C15:1n-6	0.09 ± 0.02 b	0.07 ± 0.02 b	0.13 ± 0.03 ab	0.07 ± 0.02 b	0.19 ± 0.02 a	0.09 ± 0.04 b
C16:1n-9	0.52 ± 0.04 a	0.58 ± 0.02 a	0.31 ± 0.03 b	0.29 ± 0.03 b	0.30 ± 0.05 b	0.29 ± 0.06 b
C16:1n-7(E)	0.20 ± 0.01	0.19 ± 0.03	0.22 ± 0.03	0.27 ± 0.04	0.36 ± 0.04	0.43 ± 0.04
C16:1n-7	4.93 ± 0.12 c	4.14 ± 0.35 d	4.16 ± 0.13 d	4.73 ± 0.15 cd	6.53 ± 0.29 b	7.28 ± 0.37 a
C17:1n-7	0.53 ± 0.01 bc	0.59 ± 0.05 ab	0.50 ± 0.02 bc	0.48 ± 0.02 c	0.54 ± 0.05 bc	0.65 ± 0.04 a
C18:1n-7	15.79 ± 0.68 d	15.56 ± 0.75 d	17.63 ± 0.52 cd	19.50 ± 0.48 c	26.30 ± 1.09 b	29.76 ± 1.42 a
C18:1n-9	3.20 ± 0.10 ab	2.90 ± 0.04 b	2.51 ± 0.11 b	2.68 ± 0.08 b	3.45 ± 0.18 a	3.48 ± 0.08 a
ΣMUFA	26.59 ± 1.01 c	25.35 ± 1.22 c	26.20 ± 0.85 c	28.54 ± 0.88 c	38.68 ± 1.67 b	42.93 ± 2.01 a
Poly-unsaturated fatty acids (PUFAs)						
C16:2n-6	0.05 ± 0.01	0.05 ± 0.00	0.09 ± 0.03	0.11 ± 0.02	0.04 ± 0.02	0.05 ± 0.02
C16:2n-4	0.25 ± 0.01 b	0.26 ± 0.01 b	0.37 ± 0.01 a	0.39 ± 0.03 a	0.25 ± 0.03 b	0.23 ± 0.03 b
C18:2n-6	4.06 ± 0.31 b	3.79 ± 0.34 b	20.37 ± 0.52 a	21.23 ± 0.28 a	4.55 ± 0.50 b	4.34 ± 0.42 b
C18:3n-6	0.07 ± 0.00 b	0.12 ± 0.04 b	0.06 ± 0.00 b	0.05 ± 0.01 b	0.28 ± 0.04 a	0.25 ± 0.03 a
C18:3n-3	0.85 ± 0.02 b	0.89 ± 0.05 b	1.93 ± 0.13 a	2.13 ± 0.07 a	0.70 ± 0.03 b	0.63 ± 0.03 b
C20:1n-9	0.70 ± 0.01 c	0.75 ± 0.09 c	1.21 ± 0.07 a	1.40 ± 0.05 a	0.97 ± 0.10 bc	1.18 ± 0.08 ab
C20:2n-6	0.34 ± 0.14	0.31 ± 0.07	0.71 ± 0.03	0.67 ± 0.09	0.50 ± 0.05	0.38 ± 0.10
C20:3n-6	0.30 ± 0.02	0.27 ± 0.01	0.24 ± 0.03	0.23 ± 0.05	0.50 ± 0.03	0.43 ± 0.07
C20:4n-6 ARA	1.96 ± 0.21 c	2.01 ± 0.12 c	0.52 ± 0.02 d	0.58 ± 0.05 d	4.63 ± 0.45 a	3.53 ± 0.42 b
C20:5n-3 EPA	3.72 ± 0.35 ab	3.97 ± 0.13 a	3.78 ± 0.09 ab	4.05 ± 0.08 a	3.67 ± 0.17 ab	3.36 ± 0.16 b
C22:6n-3 DHA	19.51 ± 0.85 a	19.73 ± 0.81 a	13.35 ± 1.04 b	12.06 ± 0.98 b	11.54 ± 0.89 bc	9.64 ± 0.73 c
ΣPUFA	31.81 ± 1.81 b	32.15 ± 1.52 b	42.63 ± 1.78 a	42.90 ± 1.53 a	27.63 ± 2.07 c	24.02 ± 1.89 c
Σn-6	6.78 ± 00.56 c	6.55 ± 0.47 c	21.99 ± 0.57 a	22.87 ± 0.44 a	10.50 ± 0.97 b	8.98 ± 0.93 b
Σn-3	24.08 ± 1.13 a	24.59 ± 0.89 a	19.06 ± 1.07 b	18.24 ± 1.06 b	15.91 ± 1.01 c	13.63 ± 0.88 c
Σn-3/Σn-6	3.55	3.75	0.87	0.80	1.52	1.52

Different letters within a row indicate significant difference (*p* < 0.05). Σn-6: C16:2n-6, C18:2n-6, C18:3n-6, C20:2n-6, C20:3n-6, C20:4n-6. Σn-3: C18:3n-3, C20:5n-3, C22:6n-3.

**Table 3 molecules-27-04480-t003:** The contents of free amino acids (FAAs) of back and abdominal muscles in three perciform fishes (mg/100 g of wet weight).

	Humpback Grouper		Red-Spotted Grouper		Black Seabream	
	Back	Abdomen	Back	Abdomen	Back	Abdomen
Aspartic acid ^1^	2.62 ± 0.18 b	2.53 ± 0.28 b	3.87 ± 0.14 a	4.50 ± 0.26 a	2.32 ± 0.12 b	2.78 ± 0.12 b
Glutamic acid ^1^	8.22 ± 0.31 a	7.30 ± 0.18 b	5.61 ± 0.12 c	6.82 ± 0.23 b	13.79 ± 0.37 a	14.00 ± 0.24 a
Threonine ^2^	7.42 ± 0.25 c	7.12 ± 0.26 c	7.53 ± 0.27 c	9.12 ± 0.28 ab	9.64± 0.15 a	8.69 ± 0.13 b
Serine ^2^	9.44 ± 0.31 c	8.72± 0.23 cd	12.07± 0.17 b	16.46 ± 0.22 a	8.19± 0.35 cd	7.19 ± 0.53 d
Glycine ^2^	75.83 ± 1.68 b	53.26 ± 1.11 c	86.30 ± 0.89 a	71.00± 1.09 b	44.05 ± 1.32 d	38.90 ± 0.72 e
Alanine ^2^	32.37 ± 0.36 c	31.40± 0.59 cd	29.88 ± 0.66 d	39.26 ± 0.52 a	35.09 ± 0.82 b	32.16 ± 0.74 c
Arginine ^2^	8.07 ± 0.48 c	9.32 ± 1.30 c	8.01 ± 0.94 c	9.04 ± 0.69 c	17.55 ± 0.62 a	14.53 ± 0.47 b
Proline ^2^	12.90 ± 0.28 a	10.14 ± 0.28 b	13.12 ± 0.70 a	11.74 ± 0.73 ab	13.08 ± 0.89 a	12.81 ± 0.60 a
Valine ^3^	5.21 ± 0.12 b	5.27 ± 0.28 b	2.18 ± 0.25 c	2.69 ± 0.16 c	15.94 ± 0.78 a	14.16 ± 0.55 a
Methionine ^3^	3.88 ± 0.18 b	3.87 ± 0.15 b	2.43 ± 0.16 c	3.45 ± 0.29 b	8.22 ± 0.40 a	7.53 ± 0.40 a
Isoleucine ^3^	4.56 ± 0.07 b	4.60 ± 0.24 b	1.88 ± 0.23 c	2.15± 0.26 c	14.15 ± 0.64 a	12.43 ± 0.79 a
Leucine ^3^	8.33 ± 0.37 b	8.37 ± 0.42 b	2.99 ± 0.39 c	3.53 ± 0.14 c	22.60 ± 0.89 a	20.02 ± 1.08 a
Phenylalanine ^3^	4.33 ± 0.22 b	4.64 ± 0.13 b	1.76 ± 0.07 c	2.24 ± 0.22 c	14.04 ± 0.52 a	13.36 ± 0.91 a
Histidine ^3^	4.38 ± 0.12 b	4.36 ± 0.14 b	3.63 ± 0.26 b	4.42 ± 0.24 b	9.44 ± 0.63 a	9.14 ± 0.59 a
Cysteine ^4^	6.05 ± 0.49	5.41 ± 0.27	5.06 ± 0.32	5.78 ± 0.27	6.56 ± 0.47	5.98 ± 0.16
Lysine ^4^	19.41 ± 0.25 c	18.08 ± 0.20 c	16.32 ± 0.45 d	20.44 ± 1.03 c	55.37 ± 3.39 a	46.82 ± 1.05 b
Tyrosine ^4^	3.95 ± 0.07 b	4.08 ± 0.10 b	2.57 ± 0.22 c	2.91 ± 0.19 c	11.44 ± 0.86 a	10.53 ± 0.84 a
Total FAA	216.97 ± 5.58 c	188.47 ± 5.29 d	205.21 ± 5.90 cd	215.55 ± 6.64 c	301.59 ± 10.97 a	271.03 ± 9.09 b
MSG-like FAA	10.84 ± 0.49 bc	9.83 ± 0.46 cd	9.48 ± 0.26 d	11.32 ± 0.49 b	16.11 ± 0.49 a	16.78 ± 0.36 a
Sweet FAA	146.03 ± 2.78 b	119.96 ± 3.01 cd	156.91 ± 3.13 a	156.62 ± 3.35 a	127.72 ± 3.65 c	114.28 ± 3.02 d
Bitter FAA	30.69 ± 0.98 c	31.11 ± 1.14 c	14.87 ± 1.27 d	18.48 ± 1.13 d	84.39 ± 3.76 a	76.64 ± 4.07 b
Tasteless FAA	29.41 ± 0.75 c	27.57 ± 0.57 c	23.95 ± 0.78 c	29.13 ± 1.37 c	73.37 ± 4.36 a	63.33 ± 1.96 b

Different letters within a row indicate significant difference (*p* < 0.05). ^1^ MSG-like FAAs; ^2^ Sweet FAAs; ^3^ Bitter FAAs; ^4^ Tasteless FAAs.

**Table 4 molecules-27-04480-t004:** The contents of 5′-nucleotides (mg/100 g of wet weight) of back and abdominal muscles in three perciform fishes.

	Humpback Grouper		Red-Spotted Grouper		Black Seabream	
	Back	Abdomen	Back	Abdomen	Back	Abdomen
GMP	1.04 ± 0.09 b	1.08 ± 0.07 b	0.65 ± 0.04 b	0.68 ± 0.04 b	2.81 ± 0.14 a	2.40 ± 0.36 a
IMP	302.34 ± 12.77 b	315.80 ± 18.66 b	317.90 ± 18.70 b	296.61 ± 19.68 b	371.58 ± 18.30 a	288.50 ± 19.51 b
AMP	10.20 ± 1.03 a	12.22 ± 2.34 a	0.91 ± 0.24 c	0.95 ± 0.44c	6.07 ± 1.57 b	6.42 ± 0.92 b
Total	313.58 ± 13.73 b	329.10 ± 20.26 b	319.46 ± 18.15 b	298.24 ± 19.24 b	379.83± 19.37 a	297.32 ± 20.51 b

Different letters within a row indicate significant difference (*p* < 0.05).

**Table 5 molecules-27-04480-t005:** The concentrations of organic acids and mineral ions of back and abdominal muscles in three perciform fishes.

	Humpback Grouper		Red-Spotted Grouper		Black Seabream	
	Back	Abdomen	Back	Abdomen	Back	Abdomen
**Organic acids**						
^1^ Malic acid	0.45 ± 0.06 b	0.59 ± 0.02 a	0.19 ± 0.01 d	0.33 ± 0.02 c	0.17 ± 0.03 d	0.18 ± 0.03 d
^1^ Lactic acid	3.18 ± 0.09 d	3.44 ± 0.05 d	4.90 ± 0.04 c	5.26 ± 0.14 c	7.95 ± 0.37 a	6.35± 0.19 b
**Mineral ions**						
^1^ Na^+^	0.63 ± 0.04 bc	0.57 ± 0.06 cd	0.47 ± 0.05 d	0.67 ± 0.05 bc	1.09 ± 0.04 a	0.72 ± 0.04 b
^1^ K^+^	4.40 ± 0.19 ab	4.79 ± 0.15 a	4.35 ± 0.15 ab	3.93 ± 0.11 b	4.35 ± 0.12 ab	4.64 ± 0.18 a
^1^ Ca^2+^	0.54 ± 0.11	0.39 ± 0.06	0.37 ± 0.03	0.49 ± 0.14	0.40 ± 0.06	0.33 ± 0.05
^1^ Mg^2+^	0.34 ± 0.03	0.34 ± 0.03	0.30 ± 0.02	0.30 ± 0.02	0.33 ± 0.03	0.32 ± 0.02
^2^ Mn^2+^	0.02 ± 0.01	0.02 ± 0.00	0.02 ± 0.00	0.04 ± 0.01	0.02 ± 0.01	0.02 ± 0.01
^2^ Zn^2+^	0.52 ± 0.13	0.39 ± 0.06	0.40 ± 0.02	0.50 ± 0.08	0.44 ± 0.02	0.37 ± 0.05
^1^ Cl^−^	0.56 ± 0.08 c	0.54 ± 0.03 c	0.53 ± 0.08 c	0.70 ± 0. 03 bc	0.78 ± 0.08 b	1.34 ± 0.08 a
^1^ PO_4_^3−^	6.12 ± 0.35 bc	4.86 ± 0.26 d	6.76 ± 0.48 ab	5.48 ± 0.18 cd	7.01 ± 0.13 a	6.45 ± 0.27 ab

Different letters within a row indicate significant difference (*p* < 0.05). ^1^ mg/g of wet weight; ^2^ mg/100 g of wet weight.

## Data Availability

The data presented in this study are available upon request.

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
