# Peer review of "Comparison of Biochemical Composition and Non-Volatile Taste Active Compounds of Back and Abdominal Muscles in Three Marine Perciform Fishes, Chromileptes altivelis, Epinephelus akaara and Acanthopagrus schlegelii"

_molecules, 2022, doi:10.3390/molecules27144480_

Round 1

Reviewer 1 Report

The work deals with an interesting topic of Comparison of biochemical composition and non-volatile taste 2 active compounds of back and abdominal muscles in three ma-3 rine perciforms fishes, Chromileptes altivelis, Epinephelus 4 akaara and Acanthopagrus schlegelii.
Unfortunately, it cannot be published in its current form. Notes are listed below:

-page 4, 6, 9 - the text needs to be justified.
-line 291: the authors cite reference number 44, which is not included in the publication.
-chapter 3.3. Fatty acid analysis; Information about sample preparation is missing. Information about GC analysis is missing: temperature of the injector, amount of the injected sample, injector operating mode - split or splitless (a value of split).
-chapter 3.4. Free amino acid (FAA) analysis. Information about HPLC analysis is missing: amount of the injected sample, mobile phase, detector.
-chapter 3.8. Inorganic ions assay. Information about amount of the injected sample is missing.

Author Response

Reviewer 1:

Question 1: page 4, 6, 9 - the text needs to be justified.

Respond: Thanks, the format has been revised.

Question 2: line 291: the authors cite reference number 44, which is not included in the publication.

Respond: I am sorry, this reference has been added. Furthermore, the order number of this reference was also corrected.

  1. AOAC. Official methods of analysis (14th ed.). Washington (DC): Association of Official Analytical Chemists, 1984.

Question 3: chapter 3.3. Fatty acid analysis; Information about sample preparation is missing. Information about GC analysis is missing: temperature of the injector, amount of the injected sample, injector operating mode - split or splitless (a value of split).

Respond: According to the reviewer’s advice, more information about the fatty acid analysis were added as follow “sample volume, 1 μl; split injection, 2:1; injector temperature, 260 °C”.

Question 4: chapter 3.4. Free amino acid (FAA) analysis. Information about HPLC analysis is missing: amount of the injected sample, mobile phase, detector.

Respond: Thanks. More information about HPLC analysis was added in “Free amino acid (FAA) analysis” part. In detail, “Samples were then analyzed for FAAs by a Waters 2996 high-performance liquid chromatography (HPLC) (Waters Corporation, Milford, MA, USA) equipped with a Waters 2996 Photodiode Array Detector. Samples (5 μl) were separated using a Waters Pico-Tag-C18 column (3.9 mm × 150 mm). The mobile phase consisted of AccQ-Tag Elent A (v/v=10:1) (A), 100% acetonitrile (B) and deionized water (C). Gradient elution was performed as follows: 100% A to 99% A and 1% B at 0.5 min, to 94% A and 6% B at 18 min, to 90% A and 10% B at 19 min, to 83% A and 17% B at 29.5 min, to 60% B and 40% C at 33 min, to 100% A at 36 min, and to 100% A at 45 min. The detection wavelength is 248 nm.

Question 5: chapter 3.8. Inorganic ions assay. Information about amount of the injected sample is missing.

Respond: More information about the metal ions are added as follow “Approximately 0.5 g of dried sample was digested with a reagent mixture of 1 ml of H2O2 and 5 mL of HNO3, and then submitted to a heating program in a closed microwave oven following the steps: 1 min at 320 W, 2 min at 0 W, 5 min at 320 W, 5 min at 520 W and 5 min at 740 W. The digested solution was diluted with deionized water to 100 ml and analyzed by inductively coupled plasma optical emission spectrometer.

Reviewer 2 Report

The manuscript is well written and organized. The methods and statistical analysis are scientifically sound, as well as results are properly discussed.  However. I have two main comments to the authors:

- I suggest authors to stress the novelty of their work compared to existing studies;

- It would be beneficial to expand the conclusion section, adding indication for producers from authors' results. 

- Also, conclusions have to be expanded by pointing out the limitation of the work and describing future research areas.

Author Response

Question 1: I suggest authors to stress the novelty of their work compared to existing studies.

Respond: Thanks. According to the reviewer’s advice, some relevant research reports have been added in the third paragraph of “introduction” part. In detail, “Until now many studies have focused on the effects of different dietary sources on the growth performance and biochemical composition of juvenile HG, RG and BS [5,10-12]. For example, Shapawi et al. [13] ananlyzed the fatty acid composition of juvernie HG (~10.6 g) after feeding with diets containing different vegetable oils; Wang et al [14] determined the effects of dietary protein and lipid levels on growth performance and body composition of juvenile RG (~7.85 g); Jin et al. [12] reported the effects of different dietary lipid sources on physiological characteristics and lipid metabolism in juvenile BS (~5.4 g).

Question 2: It would be beneficial to expand the conclusion section, adding indication for producers from authors' results. 

Respond: Indication for producers and customers from authors’ results has been added in “conclusion” part as follow “These findings clearly showed the differences in nutritional composition and non-volatile compounds of three adult perciforms fishes, in which HG could provide more healthy omega-3, and BS could provide stronger umami flavor taste for human.

Question 3: Also, conclusions have to be expanded by pointing out the limitation of the work and describing future research areas.

Respond: According to the reviewer’s advice, the limitation of the work and describing future research areas have been added as follow “Furthermore, besides the above parameters, other quality traits, such as texture characteristics, surface color and volatile flavor, are also important factors for consumers’ acceptability of fish meat that need to be further studied”.

Round 2

Reviewer 2 Report

I am currently fine with the current version of the manuscript . Authors properly addressed my comments.

This manuscript is a resubmission of an earlier submission. The following is a list of the peer review reports and author responses from that submission.

Round 1

Reviewer 1 Report

The research work entitled " Comparison of biochemical composition and flavor compounds of back and abdominal muscles in three marine perciforms fishes, Chromileptes altivelis, Epinephelus akaara and Acanthopagrus schlegelii”. In general, the work has been well presented, however it requires some corrections outlined below:

Regarding the abstracts, the authors presented the results obtained, however it would be interesting to present some partial conclusions from the results obtained.

In the introduction the authors report that there are few studies related to the flavor and nutritional quality of these fish, however, several parameters can influence the flavor, I believe the authors could add some studies in the introduction.

Table 1 lacks the statistical analysis of Gray.

In Table 2, why did the authors not perform statistical analysis for all fatty acids?

Did the authors find any studies that justify the predominance of glycine in some species and lysine in others?

According to the authors, lactic acid can provide a sour and umami taste in these species. Is there any information in the literature that defines a minimum concentration of lactic acid to influence the taste of fish? In relation to malic acid, what is its influence on the flavor of these species?

In item 2.5 Comparison of inorganic ions, what the authors define as “assay methods” (Page 9, Line 241).

Page 9, Lines 245 – 246, “The EUC values ​​of three perciforms fishes were calculated according to the equation used to evaluate the results of sensory evaluation (Table 4).” Present this information in the material and methods.

Did the authors perform sensory evaluation? Why didn't they show the results?

In the methods, it would be interesting for the authors to present the number of repetitions (triplicate or duplicate) for all analyses.

The conclusion needs to be changed, in general the authors only present the results presented previously, but do not draw conclusions about the manuscript, that is, what is the purpose of these characterizations? Is there a more suitable species for consumption?

Author Response

Question 1. Regarding the abstracts, the authors presented the results obtained, however it would be interesting to present some partial conclusions from the results obtained.

Responds: Thanks, some changes has been done and a short conclusion also has been added at the end of the abstract. In detail, “The total healthy omega-3 fatty acid (Σn-3) profiles in HG were the highest (24.08%-24.59%), followed by RG (18.24%-19.06%) and BS (13.63%-15.91%) (P < 0.05).”; “In conclusion, HG and RG could provide more protein and health omega-3 fatty acids than BS, while BS had stronger umami taste according to EUC values.

Question 2. In the introduction the authors report that there are few studies related to the flavor and nutritional quality of these fish, however, several parameters can influence the flavor, I believe the authors could add some studies in the introduction.

 Responds: According to the reviewer’s advice, some flavor compositions that influence the fish taste has been added in the “Introduction” part at line 64-67 “For example, umami is a specific taste of seafood, mainly caused by umami amino acids and 5’- nucleotides; ogainic acids provide acidic taste and other flavors; inorganic ions are essential auxiliary flavor components in seafood [13,14]”.

Question 3. Table 1 lacks the statistical analysis of Gray.

 Responds: Thanks, there were no significant differences among these six groups in ash contents, therefore, no letters were added in this row.

Question 4. In Table 2, why did the authors not perform statistical analysis for all fatty acids?

 Responds: This question was the same as question 3, that the data of fatty acids without letters meant no significant differences.

Question 5. Did the authors find any studies that justify the predominance of glycine in some species and lysine in others?

 Responds: Yes. As shown in line 158-161, the content of glycine was the highest in large yellow croaker and hybrid grouper reported by Wei et al.(2019) and Wang et al. (2020), while in fresh gilthead sea bream (Mente et al., 2021), the highest content of FAA was lysine, which showed the same as BS. Accordingly, this sentence has been rewritten and the new references has been added to the manuscript.

Wei, Y.T.; Shen, H.H.; Xu, W.Q.; Pan, Y.; Chen, J.; Zhang, W.B.; Mai, K.S. Replacement of dietary fishmeal by Antarctic krill meal on growth performance, intestinal morphology, body composition and organoleptic quality of large yellow croaker Larimichthys crocea. Aquaculture 2019, 512, 734281.

Wang, Z.; Qian, X.Q.; Xie, S.Q.; Yun, B. Changes of growth performance and plasma biochemical parameters of hybrid grouper (Epinephelus lanceolatus ♂ × Epinephelus fuscoguttatus ♀) in response to substitution of dietary fishmeal with poultry by-product meal. Aquacult. Rep. 2020, 18, 100516.

Mente, E.; Carter, C.G.; Barnes, R.S.; Vlahos, N.; Nengas I. Post-Prandial Amino Acid Changes in Gilthead Sea Bream. Animals 2021, 11, 1889.

Question 6. According to the authors, lactic acid can provide a sour and umami taste in these species. Is there any information in the literature that defines a minimum concentration of lactic acid to influence the taste of fish? In relation to malic acid, what is its influence on the flavor of these species?

Question 7. In item 2.5 Comparison of inorganic ions, what the authors define as “assay methods” (Page 9, Line 241).

 Responds: The assay methods has been defined in brackets as “for example, mercury thiocyanate and molybdemum blue methods were used in Cl- and PO43- were detection in some studies

Question 8. Page 9, Lines 245 – 246, “The EUC values ​​of three perciforms fishes were calculated according to the equation used to evaluate the results of sensory evaluation (Table 4).” Present this information in the material and methods.

 Responds: As shown in “3.6 . Equivalent umami concentration (EUC), line 384-395, the calculation method of EUC based on this equation, which has been widely used in many reports.

For example:

  1. Liu, Y.; Zhang, C.; Chen, S. Comparison of active non-volatile taste components in the Viscera and Adductor Muscles of Oyster (Ostrea rivularis Gould). Food Sci. Technol. Res. 2013, 19, 417–
  2. Yue, J.; Zhang, Y.F.; Jin, Y.F.; Deng, Y.; Zhao, Y.Y. Impact of high hydrostatic pressure on non-volatile and volatile compounds of squid muscles. Food Chem. 2016, 194, 12–19.
  3. Liu, C.; Gu, Z.; Lin, X.; Wang, Y.; Wang, A.; Sun, Y.; Shi, Y. Effects of high hydrostatic pressure (HHP) and storage temperature on bacterial counts, color change, fatty acids and non-volatile taste active compounds of oysters (Crassostrea ariakensis). Food Chem. 2022, 372, 131247.

Question 9. Did the authors perform sensory evaluation? Why didn't they show the results?

 Responds: In this study, we did not perform sensory evaluation according to consumer acceptance evaluation or electronic tongue analysis. And the EUC analysis [according to the contents of MSG-like amino acids (Asp and Glu) and 5′-nucleotides (IMP, GMP and AMP)] was done to evaluate the umami concentration, and results were shown in Table 4.

Question 10. In the methods, it would be interesting for the authors to present the number of repetitions (triplicate or duplicate) for all analyses.

Responds: Thanks, the number of repetitions (triplicate) were supplied in all test indexes.

Question 11. The conclusion needs to be changed, in general the authors only present the results presented previously, but do not draw conclusions about the manuscript, that is, what is the purpose of these characterizations? Is there a more suitable species for consumption?

 Responds: Thanks, the conclusion has been rewritten as follow, “The biochemical composition and flavor compounds of back and abdominal muscles in three marine perciforms fishes were compared. The crude protein contents in HG and RG were significantly higher than that in BS (P < 0.05). Though there were no significant differences in lipid contents, the profiles of healthy omega-3 fatty acids in HG were the highest, followed by RG and BS. According to the EUC values, BS back muscle showed the best flavor taste, followed by its abdomen muscle, HG back and abdomen muscles, and RG back and abdomen muscles (P < 0.05). Aliphatic hydrocarbons and aldehydes were the top two class volatile compounds in the muscles of three perciforms fishes, in which the profiles of aliphatic hydrocarbons in HG and RG were significantly higher than than in BS (P < 0.05), while the opposite tendency was observed in aldehydes (P < 0.05). In conclusion, HG could provide more healthy omega-3, and BS could provide stronger umami flavor taste for human.

Reviewer 2 Report

Thank you for letting me read this interesting study. The manuscript is carefully prepared, and well organized. However, the manuscript needs to be improved prior to publication. I would like to see the following minor issues to be addressed:

  1. Line 43 The acronym (RMB) should be defined.
  2. Line 141 Table 2 - Report unit of measure for fatty acids
  3. Line 170 All acronyms (i.e., IMP, AMP and GMP) must be defined.
  4. Line 209 The acronym (TVA) should be defined.
  5. Line 244 The acronym (EUC) should be defined on first mention.
  6. Line 253 The acronym (MSG) should be defined.
  7. Line 289 In addition to the factors listed by the authors, ketones in fish may be formed by the alcohol-dehydrogenating activity of Gram-negative bacteria
  8. Line 326 “ Crude protein content was determined by Kjeldahl method”, authors should add the citation (e.g., AOAC procedures?).
  9. What is the translational scope of this study in humans? Please define this matter into the conclusion section.
  10. Although the manuscript generally reads very well, closer inspection has raised some concerns for recommending careful re-reading by the authors to take care of some minor editorial blemishes including grammar, punctuation, spelling, space, misplaced words and improvement of overall readability.

Author Response

Question 1. Line 43 The acronym (RMB) should be defined.

Responds: The full name has been added in line 46.

Question 2. Line 141 Table 2 - Report unit of measure for fatty acids

Responds: Thanks, the unit (%) of fatty acids has been added in the Table 2.

Question 3. Line 170 All acronyms (i.e., IMP, AMP and GMP) must be defined.

Responds: The full names of IMP, AMP and GMP has been added in line 177-179.

Question 4. Line 209 The acronym (TVA) should be defined.

Responds: Sorry, TAV is correct. Accordingly the full name of TAV has been added in line 217.

Question 5. Line 244 The acronym (EUC) should be defined on first mention.

Responds: Thanks, the full name has been added.

Question 6. Line 253 The acronym (MSG) should be defined.

Responds: Thanks, the full name has been added.

Question 7. Line 289 In addition to the factors listed by the authors, ketones in fish may be formed by the alcohol-dehydrogenating activity of Gram-negative bacteria

Responds: According to the reviewer’s advice, this sentence has been changed as follow “Ketones in fish species are considered to be as a result of lipid oxidant, amino acids degradation, Maillard reactions, and alcohol-dehydrogenating activities by Gram-negative bacteria [36,42]

Question 8. Line 326 “ Crude protein content was determined by Kjeldahl method”, authors should add the citation (e.g., AOAC procedures?).

Responds: Thanks, the reference has been added.

Question 9. What is the translational scope of this study in humans? Please define this matter into the conclusion section.

Responds: Thanks. The conclusion has been rewritten, and the differences of nutrition and taste among three fish species has been evaluated. In detail, “The biochemical composition and flavor compounds of back and abdominal muscles in three marine perciforms fishes were compared. The crude protein contents in HG and RG were significantly higher than that in BS (P < 0.05). Though there were no significant differences in lipid contents, the profiles of healthy omega-3 fatty acids in HG were the highest, followed by RG and BS. According to the EUC values, BS back muscle showed the best flavor taste, followed by its abdomen muscle, HG back and abdomen muscles, and RG back and abdomen muscles (P < 0.05). Aliphatic hydrocarbons and aldehydes were the top two class volatile compounds in the muscles of three perciforms fishes, in which the profiles of aliphatic hydrocarbons in HG and RG were significantly higher than than in BS (P < 0.05), while the opposite tendency was observed in aldehydes (P < 0.05). In conclusion, HG could provide more healthy omega-3, and BS could provide stronger umami flavor taste for human.

Question 10. Although the manuscript generally reads very well, closer inspection has raised some concerns for recommending careful re-reading by the authors to take care of some minor editorial blemishes including grammar, punctuation, spelling, space, misplaced words and improvement of overall readability.

Responds: Thanks, this manuscript has been re-read, and some mistakes has been corrected.

Reviewer 3 Report

The text requires formatting in accordance with the template.

Line 11 absence: “Correspondence”.

Page 4, 5, 6, 9 - the text needs to be justified.

Table 6 is horizontal, it should be vertical. I propose to divide table 6 into 3 separate ones showing the results for: Humpback grouper, Red-spotted grouper, Black seabream.

The text also contains many punctuation errors.

In Table 2 retention indexes and units for the determined chemical compounds should be included.

Nomenclature of compounds is used incorrectly from IUPAC (Table 6, chapter 2.7). For example, is 2-hexane, 3,5,5-trimethyl-, should be 3,5,5-trimethyl-2-hexane.

In Table 6 retention indexes and units for the determined chemical compounds should be included.

In chapter 3.3 there is no description of the sample preparation and the methodology for the determination of fatty acids. The description should be similar to that in chapters 3.4 and further, where the authors also refer to the methodology described in another publication.

Chapter 3.9

Line 400-401: is “by headspace monolithic 400 material sorptive extraction-gas chromatography-mass (HS-MMSE-GC-MS)” should be: “by headspace solid phase microextraction and gas chromatography coupled with mass spectrometry (HS-SPME/GC-MS)”

Line 404: the volume of the vial should be indicated

Line 405: desorption temperature should be indicated

Line 409: is ”detector interface temperature, 280 °C” should be “transfer line temperature, 280°C”

Line 413: Authors should provide information on the basis of which MS libraries were identified the compounds. The authors also did not set retention indexes for individual compounds, which undermines the credibility of their identification.

In chapter 3.9 the authors did not provide information on how many times a sample was analyzed.

Author Response

Question 1. The text requires formatting in accordance with the template.

Responds: Thanks, the text has been formatted.

Question 2. Line 11 absence: “Correspondence”.

Responds: The mistake has been corrected.

Question 3. Page 4, 5, 6, 9 - the text needs to be justified.

Responds: The mistakes have been corrected.

Question 4. Table 6 is horizontal, it should be vertical. I propose to divide table 6 into 3 separate ones showing the results for: Humpback grouper, Red-spotted grouper, Black seabream.

Responds: Table 6 has been changed to be vertical.

Question 5. The text also contains many punctuation errors.

Responds: Thanks, these mistakes has been corrected.

Question 6. In Table 2 retention indexes and units for the determined chemical compounds should be included.

Responds: According to the reviewer’s advice, the retention time and units of fatty acid profiles has been added in Table 2.

Question 7. Nomenclature of compounds is used incorrectly from IUPAC (Table 6, chapter 2.7). For example, is 2-hexane, 3,5,5-trimethyl-, should be 3,5,5-trimethyl-2-hexane.

Responds: Thanks, all the wrong nomenclatures of compounds in Table 6 and main text were corrected.

Question 8. In Table 6 retention indexes and units for the determined chemical compounds should be included.

Responds: According to the reviewer’s advice, the unit of volatile profiles (%) has been added in Table 6. Moreover, the retention time, as well as CAS numbers of these detected volatile profiles were provided in supplementary Materials, named as Table S1.

Question 9. In chapter 3.3 there is no description of the sample preparation and the methodology for the determination of fatty acids. The description should be similar to that in chapters 3.4 and further, where the authors also refer to the methodology described in another publication.

Responds: Thanks. The method of fatty acids detection has been rewritten in line 345-357. In detail, “The fatty acids were extracted and fatty acid methyl esters (FAMEs) were prepared according to previous studies [13,23]. In detial, total lipids were extracted with chloroform–methanol (2:1, V/V), then saponified, followed by esterification, and finally extraction of FAMEs in hexane. The FAMEs were analyzed using a 6890 Hewlett-Packard gas–liquid chromatograph (Hewlett-Packard Co., Palo Alto, USA) equipped with diethylene glycol succinate column (0.25 mm i.d. × 30 m  length, 0.53 μm) (Agilent Technologies Inc., Shanghai, China) and flame ionization detector. The carrier gas was nitrogen with a flow at 1.0 mL/min. The oven temperature was maintained at 90 °C for 1 min and then increased to 180 °C at 30 °C/min, and to 210 °C at 10 °C/min and then maintained for 10 min, and then to 250 °C at 10 °C/min and maintained for 5 min. The injector and detector temperatures were all maintained at 210 °C. Quantitative data were calculated using the peak area ratio (% total fatty acids). All measurements were performed in triplicate.

Question 10. Line 400-401: is “by headspace monolithic 400 material sorptive extraction-gas chromatography-mass (HS-MMSE-GC-MS)” should be: “by headspace solid phase microextraction and gas chromatography coupled with mass spectrometry (HS-SPME/GC-MS)”

Responds: The mistake has been corrected.

Question 11. Line 404: the volume of the vial should be indicated

Responds: The vial used in this experiment was 20 mL, and has been added in line 423.

Question 12. Line 405: desorption temperature should be indicated

Responds: Thanks, the desorption temperature 280 °C has been added in the manuscript.

Question 13. Line 409: is ”detector interface temperature, 280 °C” should be “transfer line temperature, 280°C”

Responds: The mistake has been corrected.

Question 14. Line 413: Authors should provide information on the basis of which MS libraries were identified the compounds. The authors also did not set retention indexes for individual compounds, which undermines the credibility of their identification.

Responds: The MS libraries used in this study was National Institute of Standards and Technology (NIST), which has been added in manuscript. Moreover, as shown in question 6, the retention time of each compound was supplied in Table S1.

Question 15. In chapter 3.9 the authors did not provide information on how many times a sample was analyzed.

Responds: All samples were detected in triplicate, which has been added in manuscript.

Round 2

Reviewer 3 Report

The work deals with an interesting topic of analysis of biochemical composition and flavor compounds of back and abdominal muscles in three marine perciforms fishes.

Unfortunately, only some of the remarks contained in the previous review were taken into account by the publication’s authors.

The authors say that the text on pages 4, 5, 6 and 9 is justified, but they did not do it. Table 6 is still horizontal and should be changed to vertical. However, these are minor editorial errors that do not affect the works scientific value.

Unfortunately, the work also has serious factual errors and should not be published in its current form.

In tables 2 and 6, the authors added retention times instead of Kovats retention indexes, and the determined compounds cannot be verified.

In addition, the authors, at my request, added the desorption temperature of the SPME  fiber (DVB/CAR/PDMS). The value of 280oC given by them is controversial because it is higher than the maximum permissible temperature for this fiber, which is 270oC (manufacturer's specification attached to the review).

Author Response

Question 1. The authors say that the text on pages 4, 5, 6 and 9 is justified, but they did not do it. Table 6 is still horizontal and should be changed to vertical. However, these are minor editorial errors that do not affect the works scientific value.

Responds: Sorry, I don’t know what the exact error is mentioned by reviewer. Last time, we carefully checked the format error, and corrected some changes, for example the header rows in all tables were shown in bold fonts. Furthermore, as for the Table 6, its layout of paper has been changed from horizontal to vertical. If there are other mistakes in this manuscript, please point out. Many thanks.

Question 2. In tables 2 and 6, the authors added retention times instead of Kovats retention indexes, and the determined compounds cannot be verified.

Responds: Thanks. the Kovats retention indexes and retention times of both fatty acids and volatile compounds were provided. However, according to numerous published researches by our team and other authors, these parameters are often not provided in the text. Therefore, we provided these data in Table S1 and S2.

Question 3. In addition, the authors, at my request, added the desorption temperature of the SPME  fiber (DVB/CAR/PDMS). The value of 280°C given by them is controversial because it is higher than the maximum permissible temperature for this fiber, which is 270°C (manufacturer's specification attached to the review).

Responds: I am sorry that the desorption temperature is 250°C. The temperature of 280°C is “Front Injection Temperature” and “Transfer Line Temperature”. The detailed parameters of GC-MS are showed as follow:

Item

Parameter

Sample Volume

1 μL

Front Inlet Mode

8:1

Front Inlet Septum Purge Flow

3 mL min−1

Carrier Gas

Helium

Column

HP-5MS(30 m×250 μm×0.25 μm)

Column Flow

1 mL min−1

Oven Temperature Ramp

40 °C hold on 1 min; raised to160°C at a rate of 3 °C min−1, raised to 230°C at a rate of 15 °C min−1 hold on 1 min;

Front Injection Temperature

280 °C

Transfer Line Temperature

280 °C

Ion Source Temperature

230 °C

Quad Temperature

150 °C

Electron Energy

-70eV

Mass Range

m/z:50-550

Solvent Delay

2 min
